# Understanding phosphorus dynamics and uptake in diverse soils of Indo-Gangetic Plains of India

K. K. Rao [1]*, Mandira Barman[2]*, Debarup Das[2], Vinod Kumar Sharma[2], Debrup Ghosh[2], H. Krishna[2], T. K. Das[2]

1 Regional Coastal Rice Research Station, ICAR Central rice Research Institute, Srikakulam, Andhra Pradesh, 2 ICAR-Indian Agricultural Research Institute, New Delhi

* Koti1012@gmail.com (KKR); mandira.ssaciari@gmail.com (MB)

## Abstract

Phosphorus (P) is a critical macronutrient governing crop productivity, yet its availability in soils is limited by soil physico-chemical properties and interactions with other nutrients. The present study investigated phosphorus dynamics and uptake by wheat grown on twenty-five soils collected from different locations across the Indo-Gangetic Plains (IGP) of India. The soils exhibited wide variation in texture, pH (4.3–9.1), organic carbon (0.14–0.9%), and available macro and micro nutrients. Phosphate ion concentration and phosphate potential varied significantly ($p < 0.05$), reflecting contrasting P-supplying capacities. A pot experiment was conducted with wheat under control (no P) and 100% recommended P fertilizer application. Significant variation ($p < 0.05$) was observed in dry matter yield, P concentration, and P uptake among soils and treatments. Fertilizer P application enhanced dry matter accumulation by 10–70% depending on the soil, and P uptake increased more than twice in responsive soils. Particularly, P fertilization altered Zn concentration and the P/Zn ratio in wheat straw, indicating nutrient interactions. The results highlight that soil variability across the IGP governs P availability and wheat response, with implications for site-specific nutrient management to enhance productivity and minimize P-use inefficiencies.

## Introduction

Wheat (*Triticum aestivum* L.) is the second most crucial staple food crop in India, following rice, covering around 30 million hectares with yearly production surpassing 110 million tonnes [1,2]. The Indo-Gangetic Plains (IGP), extending from Punjab and Haryana in the west across Uttar Pradesh and Bihar to West Bengal in the east, forms the primary wheat-producing region of the nation, contributes over 70% of the country's total wheat production [3]. Maintaining wheat output in the IGP necessitates a thoughtful balance between resource utilization efficiency, soil health, and nutrient administration. Among the essential nutrients, phosphorus (P) is one of the vital

**Data availability statement:** All relevant data are within the paper and its Supporting Information files.

**Funding:** Part of the research was supported by a grant from the Bill & Melinda Gates Foundation (Grant 467 number # OPP1215722) sub-grant to India for the Zn mainstreaming project. The funders had no role in study design, data collection and analysis, decision to publish, or preparation of the manuscript.

**Competing interests:** The authors have declared that no competing interests exist.

nutrients that significantly impacts early crop establishment, root growth, tillering, and grain development [4].

Soil processes such as adsorption, precipitation, and coprecipitation which are significantly affected by soil mineral composition, pH levels, organic matter content, and microbial activity can affect P bioavailability, leading to limited plant growth and excessive use of P fertilizers, with adverse impacts on the environment and progressive depletion of P reserves. [4,5]. In the soils of the IGP, there is significant variability regarding texture, organic matter, and pH, leading to notable differences in P availability. In acidic soils, Fe and Al oxides usually fix P, whereas in calcareous soils, Ca–P precipitation prevails, decreasing the solubility of phosphate ions. As a result, a significant portion of applied fertilizer P quickly becomes inaccessible to plants, leading to low P use efficiency, below 20% during the initial season [6].

Phosphate potential (PP) and phosphate ion concentration has been extensively utilized to characterize the level of accessible P in soil solution [7]. The concentration of phosphate ions indicates the direct availability of P in the soil solution, whereas phosphate potential considers the soil's buffering capacity and its capability to sustain solution P as it is taken up by plants. Soils that have low phosphate potential cannot sustain sufficient P in solution, leading to significant crop reactions to fertilizer use [7,8]. In contrast, soils with elevated phosphate potential might exhibit reduced responses due to a higher natural P availability. These indices create an important structure for understanding soil P behavior beyond just basic soil test results and assist in clarifying varying crop reactions among different soils [9].

An additional aspect of phosphorus nutrition in wheat is its relationship with micronutrients, especially zinc (Zn). In soils and plants, P and Zn have an antagonistic interaction referred to as "P-induced Zn deficiency" [10]. The disparity between the supply of macronutrients (like P) and the accessibility of micronutrients (such as Zn) poses a significant challenge to attaining both high yields and nutritional quality in wheat [11]. In this context, understanding the interplay between P and Zn under varying soil conditions is crucial for achieving both nutrient efficiency and crop nutritional quality. Linking P–Zn interactions with soil properties and plant uptake dynamics provides a mechanistic foundation for optimizing fertilizer management. Therefore, this study aims to elucidate these nutrient relationships across diverse soils of the Indo-Gangetic Plains to support site-specific nutrient management strategies.

Earlier research on P availability in the IGP has mainly concentrated on soil test P and the responses in crop yields [12]. Nevertheless, these methods frequently do not reflect the dynamic characteristics of soil P and its relationship with crop needs. Using phosphate ion concentration and phosphate potential as indicators offers a deeper mechanistic insight into P availability. When paired with data on plant uptake and growth, these indices can assist in pinpointing soils that show a strong response to P application compared to those that exhibit a weaker response.

Considering the increasing challenges of sustaining wheat productivity under conditions of nutrient imbalance and environmental stress, a systematic evaluation of P dynamics in relation to soil properties and wheat response is urgently needed. The Indo-Gangetic Plains provide an ideal setting for this investigation because of their

agro ecological diversity, intensive cropping, and central role in India's food security. In this context, the present study was undertaken (i) To characterize the physico-chemical properties of representative soils from the Indo-Gangetic Plains in relation to P availability (ii) To quantify phosphate ion concentration and phosphate potential of the soils and assess their variability and (iii) To evaluate the growth, P uptake, and nutrient interactions of wheat grown on these soils under control and phosphorus-fertilized conditions.

## Materials and methods

### Location and collection of soil samples

To ensure variation in various physical and chemical properties of soil, including P status, soil samples were collected from 25 locations of Indo Gangetic Plains of India with varying pH and soil physico-chemical properties (S1 Table in S1 File). The collected soil samples were first air-dried and then ground well. Then it was passed through 2 mm sieve, and a part of the processed soil samples was used for characterization of different initial physico-chemical properties viz., texture [13], pH [14], organic carbon [15], available nitrogen [16], available phosphorus [17,18], available potassium [19] and DTPA extractable zinc (Zn) content [20] using standard analysis protocols, while another part was used for green-house pot experiment.

No specific permits were required for collecting soil samples and conducting this study because all soil samples were collected from farmers' fields with their prior consent. The study did not involve any protected areas, restricted zones, endangered species, or activities requiring governmental or institutional permits. Field access and sampling were carried out only after obtaining verbal permission from the respective landowners.

### Phosphate potential

The calculations for phosphate potential (PP) are given below as described by Aslyng [21]:

$$\text{Phosphate Potential (PP)} = 1/2\, \text{pCa} + \text{pH}_2\text{PO}_4 \tag{i}$$

$$1/2\, \text{pCa} = -1/2\, (\log C_{Ca} + \log f_{Ca}) \tag{ii}$$

Where, $C_{Ca}$ = 0.01 M

$$\log f_{Ca} = \frac{0.5\, z^2\, \sqrt{\mu}}{1 + 1.5\sqrt{\mu}} \tag{iii}$$

Where,
z = Valency of ion; $\mu$ = ionic strength

$$\mu = 1/2 \sum C_i Z_{i^2} \tag{iv}$$

Where, C= $H_2PO_4^-$ ion concentration

$$\text{pH}_2\text{PO}_4 = \text{pC}_{H2PO4} + \text{p}^{fH_2PO_4} \tag{v}$$

$$\text{pH}_2\text{PO}_4 = -\log(\log C_{H_2PO_4} + \log f^{H_2PO_4}) \tag{vi}$$

$$pC_{H_2PO_4} = pP + P(H^+/K^{II} + H^+)$$

(vii)

Where,

P = total concentration of phosphorus in the solution

$P(H^+/K^{II}+H^+)$ = correction factor derived by Aslyng [21] which relates the ratio $H_2PO_4/P$ to pH.

$K^{II}$ = second ionization constant of $H_2PO_4^-$ i.e., $10^{-7.2}$

pP = -log (P concentration)

$$\log f_{H_2PO_4} = \frac{0.5\, z^2\, \sqrt{\mu}}{1 + 1.5\sqrt{\mu}}$$

(viii)

## Greenhouse pot experiment

The greenhouse experiment was conducted in pots using all 25 experimental soils to assess the P uptake by plant using wheat (*Triticum aestivum* L) as a test crop. Three kg of soil was poured into plastic pots. A uniform basal dose (150:60) of N and $K_2O$ was added to the soil of each pot through urea and muriate of potash, respectively. All the NK fertilizers were added in solution form and thoroughly mixed with soil. Soil samples collected from 25 locations received 2 treatments (100% P and no P [control]) and each treatment combination was replicated thrice in a completely randomized design using 150 pots. P fertilizer added through Diammonium phosphate in the treated pots. The soil in each pot was then irrigated to field capacity with deionized water and incubated for one week at ambient temperature. Fifteen wheat seeds (variety HD 3086) were sown and after two weeks of sowing, a uniform plant population (five plants pot$^{-1}$) was maintained in each pot. Pots were watered daily with the required amount of water on weight basis to maintain field capacity. The plants were harvested at the flowering stage, *i.e.*, after 65 days of sowing.

## Analysis of plant samples

The harvested above ground plant samples were then washed first under a running tap water, followed by rinsing with dilute HCl (0.001 *N*), distilled water and finally with deionized water. Washed samples were first air-dried. After that, they were kept in hot air oven at 65 ± 2 °C. When the samples attained constant weight, dry biomass yield of the plant samples were recorded and further they were ground in a mechanical grinder made of stainless steel. Ground samples were stored in paper packets for further analysis. Plant samples were digested using a di-acid mixture ($HNO_3$:$HClO_4$::10:4). 0.5 g ground plant sample was taken in 150 mL conical flask and it was predigested with 10 mL of concentrated $HNO_3$. Then 5 mL of di-acid mixture was added to it and the flasks were kept on hot plate as described by Piper [22]. After digestion, the material was cooled, diluted, filtered and the volume was made up to 100 mL. Total P content of the digested samples were determined by vanadomolybdo-phosphoric yellow colour method at 420 nm using a UV-VIS spectrophotometer. Zinc content of the samples was estimated using atomic adsorption spectrophotometer.

## Statistical analysis

The data generated were subjected to analysis of variance (ANOVA) technique to know significant difference among treatments. Duncan's Multiple Range Test (DMRT) test was used for multiple comparisons among the treatments at $p < 0.05$ using GRAPES (General R-based Analysis Platform for Experimental Statistics) developed by Gopinath et al. [23], Kerala Agricultural University (kaugrapes.com).

# Results

## Initial soil physico-chemical properties

Soil samples selected for the present investigation shows a wide range of variation in physical and chemical properties (Table 1 in S1 File). Texture varied considerably: sandy loam was predominant (S1, S2, S3, S7, S9, S12, S18, S19, S25), followed by sandy clay loam (S5, S8, S16, S20, S22), loam (S4, S13, S17, S21), clay loam (S14, S15, S23, S24), silt loam (S11), and clay (S10). Soil reaction showed marked diversity, ranging from strongly acidic (pH 4.3 in S20, 5.0 in S19) to strongly alkaline (pH 9.1 in S4), suggesting varying degrees of base saturation and exchangeable acidity. Oxidizable organic carbon (OC) levels were generally low, ranging from 0.14% (S5, S12) to 0.90% (S21), reflecting organic matter depletion under intensive cultivation. Available nitrogen ranged from as low as 92 kg/ha (S5) to as high as 347 kg/ha (S25), indicating large variability in soil fertility status. Phosphorus availability differed widely; the lowest values were 2.48 kg/ha (S6) and 3.45 kg/ha (S22), whereas soils S19 and S20 showed unusually high P availability (65.4 and 33.48 kg/ha, respectively), attributed to soil acidity enhancing solubilization. Potassium availability ranged between 98 kg/ha (S23) and 482 kg/ha (S2). DTPA-extractable Zn showed moderate to wide variation, from deficient levels (0.4 mg/kg in

**Table 1. Initial physico-chemical properties of selected soils of Indo Gangetic Plains.**

| S.No. | Sand (%) | Silt (%) | Clay (%) | Texture | pH | OC (%) | Available N (kg/ha) | Available P (kg/ha) | Available K (kg/ha) | DTPA-Zn (mg/kg) |
|---|---|---|---|---|---|---|---|---|---|---|
| S1 | 77.7 | 10 | 12.3 | Sandy loam | 6.7 | 0.31 | 301 | 35.1 | 291 | 1.1 |
| S2 | 74.4 | 15 | 10.6 | Sandy loam | 8.1 | 0.49 | 121 | 27.5 | 482 | 0.7 |
| S3 | 75.7 | 12 | 12.3 | Sandy loam | 8.6 | 0.20 | 100 | 10.6 | 149 | 1.2 |
| S4 | 43.7 | 34 | 22.3 | Loam | 9.1 | 0.32 | 171 | 4.50 | 183 | 1.2 |
| S5 | 58.4 | 18 | 23.6 | Sandy clay loam | 8.5 | 0.14 | 92 | 18.2 | 308 | 0.7 |
| S6 | 66.0 | 16 | 18.0 | Sandy loam | 8.6 | 0.67 | 205 | 2.50 | 350 | 1.0 |
| S7 | 71.7 | 14 | 14.3 | Sandy loam | 8.1 | 0.49 | 171 | 5.20 | 229 | 2.1 |
| S8 | 53.7 | 22 | 24.6 | Sandy clay loam | 8.3 | 0.20 | 151 | 10.2 | 352 | 1.0 |
| S9 | 53.7 | 28 | 18.3 | Sandy loam | 7.9 | 0.36 | 180 | 6.30 | 100 | 1.1 |
| S10 | 18.4 | 38 | 43.6 | Clay | 7.4 | 0.16 | 151 | 14.2 | 189 | 1.7 |
| S11 | 25.5 | 67 | 8.0 | Silt loam | 7.6 | 0.40 | 167 | 13.4 | 150 | 0.9 |
| S12 | 56.3 | 28 | 16.2 | Sandy Loam | 8.1 | 0.14 | 117 | 10.6 | 153 | 1.1 |
| S13 | 30.4 | 46 | 23.6 | Loam | 7.7 | 0.36 | 243 | 13.5 | 132 | 1.1 |
| S14 | 34.4 | 38 | 27.6 | Clay loam | 7.9 | 0.19 | 243 | 18.7 | 327 | 0.9 |
| S15 | 40.9 | 29 | 30.1 | Clay loam | 7.2 | 0.48 | 197 | 14.2 | 145 | 0.6 |
| S16 | 55.6 | 16 | 28.9 | Sandy Clay loam | 7.2 | 0.42 | 243 | 11.7 | 126 | 0.9 |
| S17 | 48.4 | 34 | 17.6 | Loam | 8.4 | 0.52 | 171 | 30.3 | 177 | 1.4 |
| S18 | 55.0 | 35 | 10.0 | Sandy loam | 8.6 | 0.15 | 159 | 9.40 | 149 | 0.9 |
| S19 | 64.4 | 24 | 11.6 | Sandy loam | 5.0 | 0.46 | 230 | 65.4* | 160 | 2.8 |
| S20 | 63.5 | 17 | 23.2 | Sandy clay loam | 4.3 | 0.69 | 280 | 33.5* | 372 | 0.9 |
| S21 | 37.0 | 40 | 23.0 | Loam | 7.9 | 0.90 | 301 | 3.50 | 155 | 0.4 |
| S22 | 56.0 | 23 | 21.0 | Sandy clay loam | 6.7 | 0.47 | 243 | 3.50 | 121 | 0.9 |
| S23 | 22.4 | 50 | 27.6 | Clay loam | 6.4 | 0.53 | 230 | 29.2 | 98 | 1.1 |
| S24 | 25.7 | 44 | 30.3 | Clay loam | 7.0 | 0.30 | 234 | 11.6 | 220 | 0.4 |
| S25 | 64.4 | 20 | 15.6 | Sandy loam | 6.3 | 0.54 | 347 | 38.7 | 346 | 0.9 |

*Bray's P-1 Phosphorus: where OC: oxidizable organic carbon; N: Nitrogen; P: Phosphorus; K: Potassium: DTPA-Zn: Diethylene Triamine Penta Acetic Acid Extractable Zinc.

S1–S25: location of sampling.

S21, S24) to adequate (2.8 mg/kg in S19). Overall, the data indicate that soils in the region differ significantly in fertility, nutrient reserves, and chemical environment, which directly influences nutrient dynamics and crop responses.

## Phosphate potential

Marked variability was observed in phosphate ion concentration and phosphate potential among the soils (Table 2). Extractable $H_2PO_4^-$ ranged from as low as 390 µM in S19 to an extremely high 3901 µM in the same soil under acidic conditions, emphasizing the strong influence of soil pH on phosphate solubility. Acidic soils such as S19 and S20 had markedly higher $H_2PO_4^-$ concentrations (3901 and 1832 µM, respectively), while alkaline soils like S6, S9, and S10 showed much lower values (442–563 µM). The phosphate potential (PP) varied from 3.55 (S19) to 4.55 (S13 and S16), indicating contrasting phosphate buffering capacities across soils. $pfH_2PO_4$ values were highest in S19 (0.0207) and S20 (0.0125), reflecting greater phosphate intensity in acidic soils, while lowest values (0.007–0.0073) were recorded in alkaline sandy loam soils (S6, S12, S16). Soils S17 and S2 also showed moderately high $H_2PO_4^-$ values (1777 and 1560 µM, respectively), while soils S4 and S9 recorded lower levels, suggesting stronger P fixation tendencies. The data demonstrate that soil pH, texture, and organic matter collectively influence phosphate intensity and buffering. Acid soils are characterized by higher P solubility but lower buffering capacity, whereas alkaline soils show stronger buffering, limiting immediate

**Table 2. Phosphate ion concentration and phosphate potential of the selected soils of Indo Gangetic plains.**

| S.No | pH (CaCl$_2$) extractable) | H$_2$PO$_4^-$ (M) (*10$^{-6}$) | pH$_2$PO$_4^-$ | CF | pfH$_2$PO$_4$ | Phosphate potential (PP) |
|------|------|------|------|------|------|------|
| S1 | 6.35 | 874 cd | 3.06def | 0.0008 | 0.0101 | 4.20def |
| S2 | 7.51 | 1560bc | 2.81gh | 0.0006 | 0.0134 | 3.95gh |
| S3 | 7.58 | 762 cd | 3.13bcdef | 0.0002 | 0.0094 | 4.27bcdef |
| S4 | 7.46 | 532d | 3.28abcd | 0.0002 | 0.008 | 4.42abcd |
| S5 | 7.28 | 1099bcd | 2.96fg | 0.0005 | 0.0113 | 4.10fg |
| S6 | 7.36 | 453d | 3.35ab | 0.0002 | 0.0073 | 4.49ab |
| S7 | 7.14 | 1244bcd | 2.91fgh | 0.0007 | 0.012 | 4.05fgh |
| S8 | 7.13 | 795 cd | 3.10cdef | 0.0004 | 0.0097 | 4.24cdef |
| S9 | 7.1 | 476d | 3.33abc | 0.0003 | 0.0075 | 4.47abc |
| S10 | 6.7 | 472d | 3.33abc | 0.0004 | 0.0075 | 4.47abc |
| S11 | 6.77 | 819 cd | 3.09def | 0.0006 | 0.0098 | 4.23def |
| S12 | 7.09 | 442d | 3.36a | 0.0003 | 0.0073 | 4.50a |
| S13 | 6.96 | 397d | 3.41a | 0.0003 | 0.0069 | 4.55a |
| S14 | 7.18 | 1021bcd | 2.99efg | 0.0005 | 0.0109 | 4.13efg |
| S15 | 6.54 | 563d | 3.26abcd | 0.0005 | 0.0081 | 4.40abcd |
| S16 | 6.52 | 409d | 3.39a | 0.0003 | 0.007 | 4.53a |
| S17 | 7.33 | 1777b | 2.75h | 0.0008 | 0.0143 | 3.89h |
| S18 | 7.41 | 587d | 3.23abcd | 0.0002 | 0.0083 | 4.37abcd |
| S19 | 4.75 | 3901a | 2.41i | 0.0039 | 0.0207 | 3.55i |
| S20 | 4.75 | 1832b | 2.99fg | 0.0018 | 0.0125 | 4.13fg |
| S21 | 6.92 | 560d | 3.25abcd | 0.0004 | 0.0082 | 4.39abcd |
| S22 | 6.66 | 620d | 3.21abcde | 0.0005 | 0.0086 | 4.35abcde |
| S23 | 5.94 | 560d | 3.25abcd | 0.0005 | 0.0082 | 4.39abcd |
| S24 | 6.92 | 434d | 3.36a | 0.0003 | 0.0072 | 4.50a |
| S25 | 5.93 | 593d | 3.23abcd | 0.0006 | 0.0084 | 4.37abcd |

Where, S1-S25: location of sampling; CF: correction factor; Mean values followed by different small letters are significantly different (P<0.05).

P availability. This highlights the complexity of phosphorus dynamics across contrasting soil environments of the Indo-Gangetic Plains, where both inherent properties and management interventions decide nutrient supply to crops.

## Dry weight and P uptake

Wheat responded variably to phosphorus application across soils, highlighting the contrasting nutrient supply capacities (Table 3). Dry weight under control ranged from just 4.07 g/pot (S19) to 14.9 g/pot (S16), with marked improvement under P fertilization, reaching up to 16.4 g/pot (S14) and 16.1 g/pot (S19, S20, S24). The relative increase in biomass due to P was particularly observed in low-P soils such as S19 (from 4.07 to 16.1 g/pot) and S24 (from 5.30 to 15.9 g/pot), indicating high fertilizer responsiveness. Phosphorus concentration in wheat plants also increased with P application, from a minimum of 0.37 mg/g (S13, control) to 3.57 mg/g (S23, fertilized). P uptake followed the same trend, with low uptake in controls (5.20 mg/pot in S13) and significant enhancement under fertilization, peaking at 54.7 mg/pot (S19) and 49.2 mg/pot (S15). Across soils, the magnitude of increase in P uptake varied widely, with some soils (S13, S24) showing more than

**Table 3. Dry weight, P concentration and uptake by wheat in selected soils of Indo Gangetic Plains.**

| Treatment | Dry weight (g/pot) | | P Concentration (mg/g) | | P Uptake (mg/pot) | |
|---|---|---|---|---|---|---|
| | Control | 100% P | Control | 100% P | Control | 100% P |
| S1 | 10.9±0.9 | 11.7±1.2 | 2.10±0.24 | 2.60±0.04 | 23.0±4.4 | 30.5±3.3 |
| S2 | 8.13±0.7 | 13.5±1.2 | 0.88±0.04 | 2.79±0.06 | 7.20±0.4 | 37.5±3.0 |
| S3 | 12.7±1.1 | 14.4±1.1 | 1.62±0.21 | 1.75±0.06 | 20.6±4.3 | 25.3±2.4 |
| S4 | 10.6±0.8 | 13.8±1.9 | 1.23±0.04 | 1.49±0.10 | 13.1±1.1 | 20.4±1.4 |
| S5 | 10.7±0.4 | 13.4±1.1 | 2.04±0.15 | 2.54±0.07 | 21.9±1.4 | 34.0±2.2 |
| S6 | 7.93±0.5 | 12.3±2.1 | 0.93±0.03 | 1.69±0.38 | 7.40±0.7 | 20.3±1.3 |
| S7 | 12.8±0.6 | 14.6±1.5 | 2.36±0.14 | 2.58±0.03 | 30.2±3.0 | 37.7±4.4 |
| S8 | 14.1±0.7 | 14.7±2.1 | 1.63±0.17 | 2.15±0.40 | 22.9±1.7 | 31.9±9.4 |
| S9 | 10.7±0.9 | 12.0±0.6 | 0.99±0.12 | 2.73±0.37 | 10.6±1.9 | 32.9±5.8 |
| S10 | 7.27±0.8 | 13.9±1.7 | 1.02±0.08 | 1.24±0.10 | 7.50±1.3 | 17.2±1.9 |
| S11 | 6.03±0.9 | 14.6±2.4 | 1.07±0.12 | 1.43±0.05 | 6.40±0.4 | 20.8±3.1 |
| S12 | 10.6±1.6 | 14.8±0.3 | 0.65±0.08 | 2.43±0.08 | 6.80±0.6 | 36.0±2.0 |
| S13 | 14.2±0.7 | 14.2±1.1 | 0.37±0.09 | 2.90±0.18 | 5.20±1.2 | 41.4±5.3 |
| S14 | 10.3±0.8 | 16.4±1.3 | 2.02±0.16 | 2.67±0.13 | 20.7±0.4 | 43.7±5.1 |
| S15 | 13.3±0.3 | 14.7±0.1 | 1.41±0.14 | 3.34±0.15 | 18.8±2.0 | 49.2±1.9 |
| S16 | 14.9±0.5 | 15.8±0.5 | 0.64±0.12 | 1.54±0.07 | 9.60±2.0 | 24.3±0.5 |
| S17 | 14.3±0.7 | 14.7±0.3 | 2.63±0.22 | 2.93±0.04 | 37.6±5.0 | 43.2±1.3 |
| S18 | 11.0±0.6 | 15.0±0.3 | 1.75±0.04 | 1.83±0.03 | 19.2±0.6 | 27.3±0.5 |
| S19 | 4.07±0.5 | 16.1±1.7 | 3.08±0.12 | 3.39±0.25 | 12.6±2.0 | 54.7±9.9 |
| S20 | 14.4±0.7 | 16.1±1.4 | 1.07±0.21 | 1.66±0.06 | 15.5±3.7 | 26.8±2.8 |
| S21 | 11.5±1.3 | 13.8±1.7 | 1.48±0.21 | 1.87±0.07 | 17.2±4.4 | 25.8±2.3 |
| S22 | 11.0±0.3 | 11.5±0.6 | 1.81±0.27 | 2.46±0.02 | 19.9±2.6 | 28.2±1.4 |
| S23 | 10.7±1.4 | 13.5±1.4 | 1.27±0.25 | 3.57±0.17 | 13.5±1.2 | 48.1±2.8 |
| S24 | 5.30±0.3 | 15.9±2.0 | 1.67±0.05 | 2.33±0.09 | 8.80±0.1 | 37.2±5.9 |
| S25 | 12.0±0.4 | 15.6±1.6 | 1.47±0.27 | 1.67±0.04 | 17.6±2.9 | 26.3±2.1 |
| CV | 7.43 | 9.87 | 10.81 | 7.09 | 15.5 | 12.4 |
| LSD | 1.31 | 2.31 | 0.26 | 0.27 | 4.00 | 6.66 |
| SEM (±) | 0.46 | 0.81 | 0.09 | 0.09 | 1.41 | 2.35 |

S1-S25: location of sampling; CV: Coefficient of variation; LSD: Least significant difference; SEM: Standard error of a mean.

fivefold increases. In general, soils with low inherent available P (S19, S23, S24) responded strongly to fertilization, while those with moderate P (S7, S8, S17) showed relatively less improvement. The data reveal that phosphorus application consistently improved plant growth and nutrient acquisition, though the efficiency of uptake was soil-specific, underlining the role of inherent soil fertility and P dynamics in influencing crop response.

Cumulative data shows clear positive effects of P fertilization on wheat growth and nutrient status (Fig 1). Across soils, P application significantly increased shoot dry matter, tissue P concentration, and total uptake compared to unfertilized controls. While the degree of response varied, low-P soils showed the largest relative improvements. The data shows that native soil P was insufficient in most soils to support optimal wheat growth. Fertilization not only enhanced biomass but also improved nutrient concentration, indicating better physiological utilization of phosphorus. The trend underscores the importance of P supplementation in maximizing crop productivity in the Indo-Gangetic Plains.

## Zn × P interaction

Zinc concentration in wheat straw showed significant response under P application. In the control, Zn concentration ranged between 5.70 mg/kg (S24) and 18.3 mg/kg (S15). With P fertilization, values shifted widely, dropping drastically in soils such as S25 (0.92 mg/kg) and S6 (4.34 mg/kg), while increasing in S15 (22.7 mg/kg). This demonstrates a complex interaction where P fertilization often diluted or antagonized Zn uptake. The P/Zn ratio showed even more variation. In controls, ratios ranged from as low as 24 (S13) to as high as 293 (S24), while with P fertilization they increased sharply, particularly in S25 (1818), indicating severe nutrient imbalance. In contrast, in soils with better Zn availability (S15, S17, S19), ratios were relatively stable (147–209). Overall, P application enhanced P nutrition but often at the cost of Zn uptake, leading to elevated P/Zn ratios. The negative P–Zn interaction arises from the formation of insoluble zinc–phosphate compounds and competitive inhibition at the root interface, which together reduce Zn uptake under high phosphorus supply. This imbalance poses a risk of inducing Zn deficiency in wheat grown on soils with marginal Zn reserves, emphasizing the need for balanced nutrient management strategies.

## Correlation

The correlation analysis revealed contrasting patterns under control and fertilized conditions (Fig 2). Under control conditions, the correlation between P uptake and dry matter yield was moderate (r = 0.42, p < 0.05), while the correlation between P uptake and tissue P concentration was weaker (r = 0.31, p < 0.05). In contrast, under P-fertilized conditions, both relationships strengthened substantially, with P uptake strongly correlated with dry matter (r = 0.82, p < 0.001) and

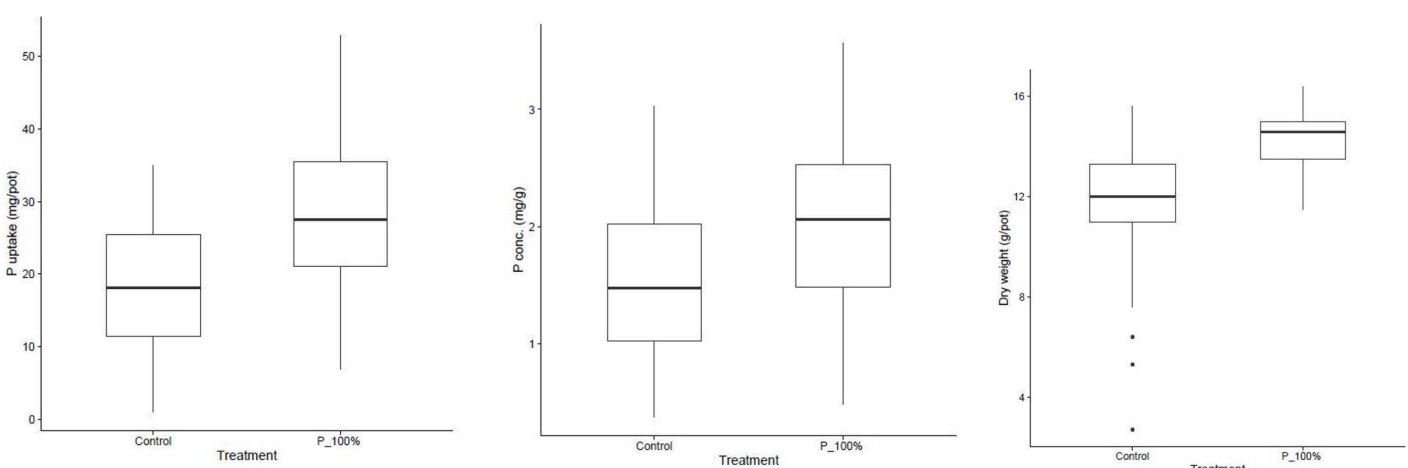

**Fig 1. Cumulative P uptake, concentration and dry weight in control and 100% P treatments.**

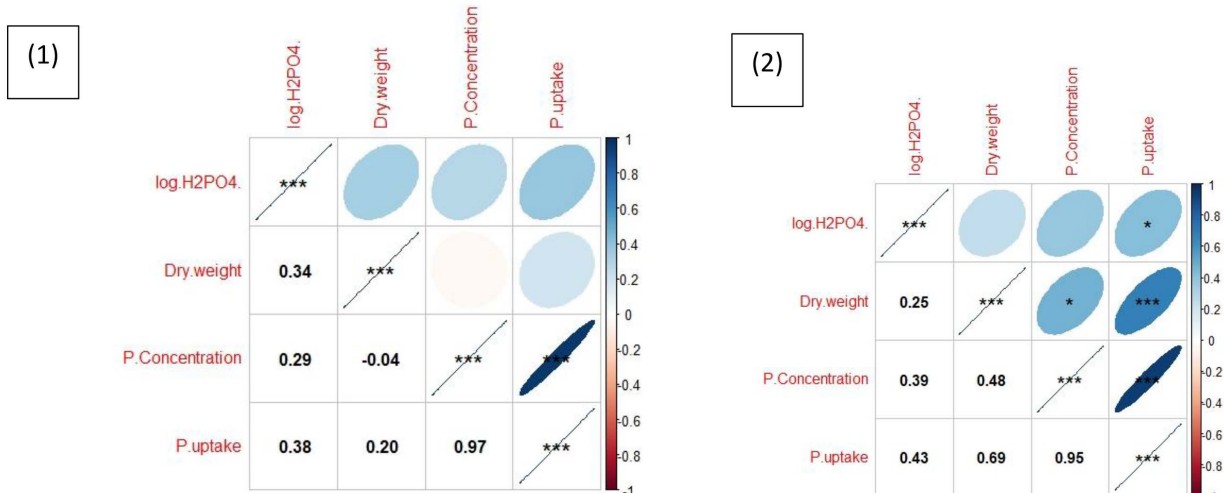

**Fig 2. Correlation matrix showing relationships between dry matter, P concentration, and P uptake under control and P-fertilized conditions (control: r=0.31–0.42, p<0.05; fertilized: r=0.76–0.82, p<0.001).**

tissue P concentration (r=0.76, p<0.001). In fertilized pots, strong positive correlations were observed between P uptake and dry matter, as well as between uptake and concentration, indicating that improved P availability drives growth and nutrient accumulation simultaneously. In contrast, under control conditions, correlations were weaker, suggesting that limited P supply constrained the relationship between growth and nutrient status. The results highlight the central role of phosphorus in determining productivity and nutrient dynamics in these soils.

## Discussion

The present study demonstrated considerable variability in the physico-chemical properties of soils from the Indo-Gangetic Plains (IGP), which in turn strongly influenced phosphorus (P) availability, wheat growth response, and the interaction with zinc (Zn). Across the 25 soils evaluated, differences in soil pH, texture, organic carbon, and baseline nutrient content were large, reflecting the well-known heterogeneity of IGP soils under intensive cultivation. Such variability has profound implications for nutrient management and crop productivity.

Soil fertility status at the outset (Table 1) indicated that organic carbon was consistently low, and that available nitrogen, phosphorus, and potassium were unevenly distributed. This is consistent with earlier studies showing the depletion of organic matter and imbalances in macro- and micronutrients in intensively cropped IGP systems [24–26]. The large spread in soil reaction (pH 4.3 to 9.1) reflects both inherent parent material and long-term management, and directly conditioned P dynamics. Acidic soils (e.g., S19, S20) exhibited unusually high extractable P, likely due to enhanced solubilization under low pH, while alkaline soils showed restricted P solubility due to precipitation with Ca. These findings align with previous reports indicating that pH is the single most important driver of phosphate solubility and fixation in IGP soils [27–29].

The analysis of phosphate ion concentrations and phosphate potential (Table 2) further reinforced this observation. Acidic soils presented higher phosphate intensity but weaker buffering capacity, whereas alkaline soils exhibited strong buffering but lower intensity. Such results highlight the inherent trade-off between solubility and buffering, which has been documented in other Indian soils [27,30]. For crop management, this means that identical rates of P fertilizer will behave differently depending on soil buffering: in low-buffer soils, applied P remains more available but is prone to leaching, while in high-buffer soils, larger applications may be required to overcome fixation. The pronounced variability in $pfH_2PO_4$ values also suggests that soils differ in their ability to maintain phosphate concentration against plant uptake, which may explain subsequent differences in crop response.

Wheat grown in pots without P fertilization had generally low biomass and nutrient uptake, confirming that native soil P was insufficient for optimal growth. Addition of 100% P fertilizer significantly improved dry matter, tissue P concentration, and total uptake (Table 3). The relative gain was most pronounced in soils with very low baseline P, such as S19 and S24, where biomass tripled or quadrupled after fertilization. This is in agreement with earlier reports that the majority of IGP soils are highly responsive to P fertilization, particularly those under rice–wheat sequences that have experienced nutrient mining [28,31,32]. The cumulative response trends (Fig 1) clearly demonstrated that P application was essential to unlock yield potential, and the correlation analysis (Fig 2) highlighted that under P sufficiency, plant growth, P concentration, and uptake were tightly coupled. In contrast, under deficiency, these relationships weakened, illustrating that P availability is the primary factor regulating the coordination between growth and nutrient accumulation.

An important dimension of the study was the antagonistic effect of P fertilization on Zn nutrition (Table 4). In many soils, P application led to reduced Zn concentration in wheat straw and a substantial increase in the P/Zn ratio. This was especially severe in soils with low baseline Zn, such as S25, where the ratio rose above 1,800. Such imbalances are of

**Table 4. Zinc (Zn) concentration and P/Zn ratio in wheat straw in selected soils of Indo Gangetic Plains.**

| Treatment | Zn Concentration (mg/kg) | | P/Zn ratio | |
|---|---|---|---|---|
| | Control | 100% P | Control | 100% P |
| S1 | 7.88±0.4 | 7.74±0.4 | 267±37 | 336±23 |
| S2 | 8.20±0.3 | 6.94±0.3 | 108±10 | 402±11 |
| S3 | 10.7±0.2 | 10.1±0.2 | 152±22 | 174±5 |
| S4 | 17.9±1.7 | 17.2±1.6 | 69±9 | 87±4 |
| S5 | 14.1±0.8 | 13.6±0.8 | 146±15 | 186±15 |
| S6 | 7.52±0.2 | 4.34±0.1 | 123±7 | 390±79 |
| S7 | 9.94±0.5 | 7.74±0.4 | 237±6 | 334±14 |
| S8 | 9.66±0.8 | 9.14±0.7 | 170±26 | 239±64 |
| S9 | 11.5±0.5 | 9.86±0.4 | 86±9 | 276±29 |
| S10 | 11.7±0.6 | 7.28±0.4 | 88±10 | 171±21 |
| S11 | 15.2±0.5 | 14.1±0.5 | 70±11 | 101±7 |
| S12 | 14.9±0.4 | 13.6±0.4 | 43±4 | 179±6 |
| S13 | 15.2±0.8 | 8.32±0.5 | 24±5 | 349±12 |
| S14 | 9.94±0.5 | 7.86±0.4 | 203±20 | 340±20 |
| S15 | 18.3±1.0 | 22.7±1.2 | 77±12 | 147±14 |
| S16 | 11.5±0.7 | 14.4±0.9 | 56±11 | 107±3 |
| S17 | 13.6±0.5 | 17.5±0.6 | 192±9 | 168±8 |
| S18 | 8.20±0.2 | 6.06±0.2 | 214±9 | 302±6 |
| S19 | 17.6±0.4 | 16.2±0.3 | 175±10 | 209±12 |
| S20 | 7.86±0.4 | 6.72±0.4 | 136±30 | 248±19 |
| S21 | 8.44±0.5 | 9.94±0.6 | 176±27 | 189±13 |
| S22 | 7.98±0.3 | 9.48±0.3 | 226±26 | 260±9 |
| S23 | 15.2±0.2 | 14.9±0.1 | 84±17 | 240±13 |
| S24 | 5.70±0.3 | 12.6±0.7 | 293±12 | 186±7 |
| S25 | 6.72±0.2 | 0.92±0.1 | 219±34 | 1818±115 |
| CV | 5.31 | 5.60 | 12.4 | 11.2 |
| LSD | 0.99 | 0.99 | 29.6 | 54.4 |
| SEM (±) | 0.35 | 0.35 | 10.4 | 19.2 |

S1-S25: location of sampling: CV: Coefficient of variation; LSD: Least significant difference; SEM: Standard error of a mean.

agronomic and nutritional concern, since high P/Zn ratios in plants are associated with hidden Zn deficiencies that may impair yield and reduce grain nutritional quality [30,33,34]. The phenomenon of P-induced Zn deficiency has been widely reported and attributed to several mechanisms: precipitation of Zn as insoluble phosphates, competitive inhibition of uptake, suppression of Zn-mobilizing mycorrhizal associations, and dilution effects due to increased biomass [34–36]. Our findings provide empirical evidence of this interaction under IGP soil conditions. Conversely, in soils with adequate Zn reserves (e.g., S15, S17), the negative impact of P was less pronounced, suggesting that Zn sufficiency buffers against P-induced stress.

These results emphasize the need for integrated P and Zn management. Previous field studies in wheat and maize have shown that combined P and Zn fertilization not only improves yield but also alleviates nutrient antagonism [5,31,37]. Co-application strategies, including Zn-enriched superphosphate or blended fertilizers, have been reported to improve nutrient use efficiency and balance P/Zn ratios [35,38]. The findings here suggest that in P-deficient yet Zn-marginal soils, applying P without Zn could exacerbate imbalances. Hence, nutrient recommendations in the IGP should avoid blanket prescriptions and instead incorporate soil test data for both macronutrients and micronutrients. The heterogeneity observed across the sampled soils supports the argument for site-specific nutrient management.

Beyond chemical fertilizers, soil organic carbon emerges as a critical factor. Across the sampled soils, organic matter was consistently low, which not only reduces nutrient retention but also limits biological processes such as mycorrhizal colonization. Conservation agriculture practices that improve organic matter levels have been shown to enhance nutrient cycling and moderate negative P–Zn interactions [26,28]. Incorporating residues, green manures, or compost could improve both P and Zn availability while enhancing soil structure and microbial health. Building organic matter would also improve phosphate buffering and reduce the volatility of nutrient responses. While P fertilization remains essential to sustain wheat productivity, the concurrent risk of Zn deficiency cannot be ignored. This is particularly relevant in the context of human nutrition, as Zn deficiency in crops translates into dietary deficiencies in populations relying heavily on wheat-based diets [30,34]. Thus, agronomic interventions addressing P–Zn interactions contribute not only to productivity but also to nutritional security.

## Conclusion

This study demonstrated that soils of the Indo-Gangetic Plains (IGP) are highly heterogeneous in their physico-chemical properties, leading to marked differences in phosphorus (P) availability, buffering capacity, and crop responsiveness. Soil pH emerged as a key driver of P dynamics, with acidic soils showing higher phosphate intensity but weaker buffering, while alkaline soils exhibited lower intensity but stronger buffering. Wheat grown in these soils responded strongly to P fertilization, particularly in low-P soils, with significant improvements in biomass, tissue P concentration, and uptake. However, P addition frequently depressed zinc (Zn) concentration in wheat straw, increasing the P/Zn ratio and indicating a risk of nutrient imbalance. The antagonistic interaction between P and Zn was most severe in soils with low baseline Zn availability, underscoring the potential for P-induced Zn deficiency under fertilization regimes that ignore micronutrient management.

These findings highlight the need for soil-specific and balanced nutrient management strategies in the IGP. Blanket recommendations of P fertilizers are unlikely to be effective and may exacerbate micronutrient deficiencies. Instead, integrated P and Zn fertilization, informed by soil testing, should be prioritized to maximize yield while safeguarding nutritional quality. Conservation agriculture practices and organic matter management could further improve nutrient buffering and enhance the biological processes that support both P and Zn uptake.

By integrating soil fertility mapping, site-specific nutrient management, and balanced fertilization strategies, sustainable wheat production in the IGP can be achieved. Such approaches will not only improve productivity but also contribute to addressing widespread micronutrient malnutrition in human populations dependent on wheat-based diets. Ultimately, managing the interaction between P and Zn is essential to reconcile crop yield goals with soil health and nutritional

security in one of the world's most intensively cultivated agricultural regions. Overall, these findings provide a scientific basis for developing site-specific nutrient management strategies that optimize phosphorus use efficiency while improving crop nutritional quality and soil health across the Indo-Gangetic Plains. These results also highlight the need for future field-based validation across diverse agro-ecological zones to refine nutrient recommendations and support long-term sustainability, ensuring that balanced P–Zn management enhances both crop productivity and environmental resilience in the Indo-Gangetic Plains.

## Supporting information

**S1 File. Location of soil sampling and raw data sets of nutrient concentration and uptake.**
(DOCX)

## Acknowledgments

This work was carried out to fulfill the requirement of PhD programme in the division of Soil Science and Agricultural Chemistry at ICAR- Indian Agricultural Research Institute, New Delhi. The authors are thankful to the Director, ICAR-IARI, Professor and faculty of the soil science division for providing laboratory facilities for conducting the experiment and analysis of soil and plant samples.

## Author contributions

**Conceptualization:** K K Rao, Mandira Barman, Debarup Das, Vinod Kumar Sharma, T.K. Das.

**Data curation:** K K Rao, Mandira Barman.

**Formal analysis:** K K Rao, Debrup Ghosh.

**Funding acquisition:** Hari Krishna.

**Investigation:** K K Rao.

**Methodology:** K K Rao, Mandira Barman, Debrup Ghosh.

**Supervision:** Mandira Barman, Debarup Das, Vinod Kumar Sharma, T.K. Das.

**Validation:** K K Rao.

**Visualization:** K K Rao, Debrup Ghosh, Hari Krishna.

**Writing – original draft:** K K Rao.

**Writing – review & editing:** K K Rao, Mandira Barman, Debarup Das, Vinod Kumar Sharma, Debrup Ghosh, Hari Krishna, T.K. Das.

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
