## [Decision Letter · Decision Letter 0]

27 Oct 2025

Dear Dr. Karnena,

Thank you for submitting your manuscript to PLOS ONE. After careful consideration, we feel that it has merit but does not fully meet PLOS ONE’s publication criteria as it currently stands. Therefore, we invite you to submit a revised version of the manuscript that addresses the points raised during the review process.

We look forward to receiving your revised manuscript.

Kind regards,

Abhay Omprakash Shirale, PhD

Academic Editor

PLOS ONE

Journal Requirements:

3. Please note that PLOS One has specific guidelines on code sharing for submissions in which author-generated code underpins the findings in the manuscript. In these cases, we expect all author-generated code to be made available without restrictions upon publication of the work. Please review our guidelines at https://journals.plos.org/plosone/s/materials-and-software-sharing#loc-sharing-code and ensure that your code is shared in a way that follows best practice and facilitates reproducibility and reuse.

4. We note that your Data Availability Statement is currently as follows: All relevant data are within the manuscript and supporting information files.

6. We note that Figure 1 in your submission contain map images which may be copyrighted. All PLOS content is published under the Creative Commons Attribution License (CC BY 4.0), which means that the manuscript, images, and Supporting Information files will be freely available online, and any third party is permitted to access, download, copy, distribute, and use these materials in any way, even commercially, with proper attribution. For these reasons, we cannot publish previously copyrighted maps or satellite images created using proprietary data, such as Google software (Google Maps, Street View, and Earth). For more information, see our copyright guidelines: http://journals.plos.org/plosone/s/licenses-and-copyright.

Reviewers' comments:

Reviewer's Responses to Questions

**Comments to the Author**

1. Is the manuscript technically sound, and do the data support the conclusions?

Reviewer #1: Yes

Reviewer #2: Yes

Reviewer #3: Yes

2. Has the statistical analysis been performed appropriately and rigorously?

Reviewer #1: Yes

Reviewer #2: Yes

Reviewer #3: Yes

3. Have the authors made all data underlying the findings in their manuscript fully available?

Reviewer #1: Yes

Reviewer #2: Yes

Reviewer #3: Yes

4. Is the manuscript presented in an intelligible fashion and written in standard English?

Reviewer #1: Yes

Reviewer #2: Yes

Reviewer #3: Yes

Reviewer #1: Comments

This manuscript addresses an important topic concerning phosphorus (P) dynamics and uptake in the Indo-Gangetic Plains (IGP), a region of high agronomic significance. The study provides valuable insights into the relationship between soil physico-chemical properties, P availability, and nutrient interactions, particularly the antagonism between P and Zn in wheat. The topic aligns well with the scope of PLOS ONE, emphasizing soil fertility, nutrient efficiency, and sustainable crop production.

However, while the study is scientifically sound and the data is extensive, the MS need some minor revisions:

Title and Abstract

• The title is appropriate; however, please correct the spelling of “Phopshorus” to “Phosphorus” in the short title.

• In the abstract, the sentence “excessive P application altered Zn concentration” may be rephrased as “P fertilization altered Zn concentration” to avoid implying over-application.

Introduction

• The introduction is well written and contextually relevant. Ensure consistent use of “phosphorus (P)” after its first occurrence.

Results

• In the Tables, please ensure consistent significant figure formatting for numeric data (e.g., 2 decimal places or as appropriate).

Discussion

• The discussion effectively relates soil variability and P dynamics and well written while focusing on the most relevant findings.

Conclusion

• The conclusion is comprehensive. You may consider a concise final sentence emphasizing the applicability of findings to site-specific nutrient management and nutritional quality improvement.

Formatting and References

• Check consistency in reference formatting (especially spacing and punctuation between author initials).

• Ensure that all abbreviations (e.g., OC, DTPA) are defined at first use.

• Correct minor typographical errors (e.g., inconsistent capitalization, spacing before parentheses).

Some specific suggestions:

• Line 92: Check grammar of this sentence.

• Line 95: Spelling mistake. Need Correction.

• Line 101: Need Capitalization.

• Line 109: this symbol (p) is not in above equation.

• Line 113, 114, 120: Write properly by using subscripts.

• Line 125: Combine it as diammonium.

• Line 127: Treatment combinations are not defined clearly, neither in a table or in-text.

• Table 1: Maintain the uniformity of digits after decimal through the table. Write expanded form of ‘Avail.’ in the footnote of the table.

• Table 2: Write expanded form of ‘CF’ and other variables in footnote of the table.

• Table 3 &4: Add expanded form and details regarding no. of observations, level of significance, etc. about CV, LSD, and SE(m) terms in table footnote.

Overall Assessment

The manuscript is well-structured, data-rich, and scientifically sound. Only minor editorial and formatting improvements are needed. The findings have clear implications for nutrient management in the Indo-Gangetic Plains and merit publication after these small revisions.

Reviewer #2: The manuscript 'Understanding phosphorus dynamics and uptake in diverse soils of Indo-Gangetic Plains of India' presents novel insights into phosphorus dynamics and nutrient interactions in wheat under diverse soil conditions. The work is generally sound and within the scope of PLOS ONE. The manuscript is well written and structured logically. The objectives are clear and align with the presented results. However, a few editorial and methodological clarifications are required. Figures and tables could benefit from consistent formatting, and certain results need quantitative support. Key issues include unclear fertilizer protocols, limited explanation of sampling design and interpretation is largely descriptive.

Line-by-Line Comments

Lines3: Correct "Phopshorus" to "Phosphorus" in the short title.

Lines 36-37: Add "wheat" and "zinc deficiency" for better discoverability.

Lines 20–35: Abstract: Condense slightly and include key quantitative findings (e.g., range of P uptake and soil pH). The abstract mentions significant variations (p<0.05) but does not specified the statistical test used in methodology section.

Lines 38–89: Introduction repeats known concepts. Condense and focus on research gap and hypothesis development.

Lines 47–56: Revise paragraph on phosphorus chemistry to include one or two recent references (2023–2024).

Lines 72–79: Briefly link the P–Zn interaction discussion with the study objectives.

Lines 147–152: Specify version and reference for statistical software (GRAPES).

Lines 90–152: Provide detailed description of soil sampling criteria, replication, and fertilizer treatments. Clarify 150% vs 100% P confusion.

Lines 175–196: Define pfH₂PO₄ and provide units in Table 2.

Lines 226–238: Provide a one-line mechanistic explanation of P–Zn antagonism for clarity.

Lines 226–238: Include two-way ANOVA results for P×Zn interaction.

Lines 242–250: Provide correlation coefficients (r, p) for Figure 3 instead of descriptive statements.

Lines 322–341: Conclusion: Add statement on implications for future field research and sustainability.

General comments

Provide correlation coefficient (r) values in Figure 3.

Ensure consistency in statistical notations across all tables

Add references to standard protocols viz. texture by hydrometer method etc..

Reviewer #3: The authors aimed to explain the dynamics of phosphorus and its uptake in various soil types across the Indo-Gangetic Plains of India. The availability of phosphorus fertilizers remains a major challenge for farmers in India. This study contributes to a better understanding of phosphorus fixation and its availability to plants.

**Do you want your identity to be public for this peer review?** For information about this choice, including consent withdrawal, please see our Privacy Policy

Reviewer #1: No

Reviewer #2: **Yes: ** gaurav mishra

Reviewer #3: **Yes: ** DR. SUSHIL KUMAR KHARIA, SKRAU, BIKANER (RAJASTHAN) INDIA

---

## [Author Response · Author response to Decision Letter 1]

17 Nov 2025

All the responses are attached as "Response to Reviews" document.

---

## [Decision Letter · Decision Letter 1]

1 Dec 2025

Understanding phosphorus dynamics and uptake in diverse soils of Indo-Gangetic Plains of India

PONE-D-25-53866R1

Dear Dr. Koteshwar Rao,

We’re pleased to inform you that your manuscript has been judged scientifically suitable for publication and will be formally accepted for publication once it meets all outstanding technical requirements.

Kind regards,

Abhay Omprakash Shirale, PhD

Academic Editor

PLOS ONE

Additional Editor Comments (optional):

Reviewers' comments:

Reviewer's Responses to Questions

**Comments to the Author**

Reviewer #1: All comments have been addressed

Reviewer #2: All comments have been addressed

Reviewer #3: All comments have been addressed

2. Is the manuscript technically sound, and do the data support the conclusions?

Reviewer #1: Yes

Reviewer #2: Yes

Reviewer #3: Yes

3. Has the statistical analysis been performed appropriately and rigorously?

Reviewer #1: Yes

Reviewer #2: Yes

Reviewer #3: Yes

4. Have the authors made all data underlying the findings in their manuscript fully available?

Reviewer #1: Yes

Reviewer #2: Yes

Reviewer #3: Yes

5. Is the manuscript presented in an intelligible fashion and written in standard English?

Reviewer #1: Yes

Reviewer #2: Yes

Reviewer #3: Yes

Reviewer #1: Thank you for sharing the revised version of the manuscript titled “Understanding Phosphorus Dynamics and Uptake in Diverse Soils of Indo-Gangetic Plains of India.” I have carefully reviewed the corrected version and am satisfied with the revisions made by the authors. The concerns raised in the previous review have been appropriately addressed, and the manuscript has improved in clarity, scientific rigor, and presentation.

I recommend accepting the manuscript in its current form.

Reviewer #2: (No Response)

Reviewer #3: The authors tried to explain phosphorus dynamics and uptake in diverse soils of Indo-Gangetic

Plains of India. Overall the paper quality is good.

**Do you want your identity to be public for this peer review?** For information about this choice, including consent withdrawal, please see our Privacy Policy

Reviewer #1: **Yes: ** Pooja Jangra

Reviewer #2: **Yes: ** GAURAV MISHRA

Reviewer #3: **Yes: ** SUSHIL KUMAR KHARIA

---

## [Editor Report · Acceptance letter]

PONE-D-25-53866R1

PLOS One

Dear Dr. Rao,

I'm pleased to inform you that your manuscript has been deemed suitable for publication in PLOS One. Congratulations! Your manuscript is now being handed over to our production team.

Kind regards,

on behalf of

Dr. Abhay Omprakash Shirale

Academic Editor

PLOS One